# The Effect of Combined Exposure of *Fusarium* Mycotoxins on Lipid Peroxidation, Antioxidant Defense, Fatty Acid Profile, and Histopathology in Laying Hens’ Liver

**DOI:** 10.3390/toxins16040179

**Published:** 2024-04-07

**Authors:** Szabina Kulcsár, Janka Turbók, György Kövér, Krisztián Balogh, Erika Zándoki, Patrik Gömbös, Omeralfaroug Ali, András Szabó, Miklós Mézes

**Affiliations:** 1Department of Feed Safety, Institute of Physiology and Nutrition, Hungarian University of Agriculture and Life Sciences, Gödöllő Campus, H-2100 Gödöllő, Hungary; balogh.krisztian.milan@uni-mate.hu; 2HUN-REN-MATE Mycotoxins in the Food Chain Research Group, Hungarian University of Agriculture and Life Sciences, H-7400 Kaposvár, Hungary; balogne.zandoki.erika@uni-mate.hu (E.Z.); szabo.andras@uni-mate.hu (A.S.); 3Agrobiotechnology and Precision Breeding for Food Security National Laboratory, Institute of Physiology and Nutrition, Department of Physiology and Animal Health, Hungarian University of Agriculture and Life Sciences, H-7400 Kaposvár, Hungary; janka.turbok@gmail.com (J.T.); gombos.patrik@uni-mate.hu (P.G.); vomer2011@gmail.com (O.A.); 4Department of Animal Science, Institute of Animal Breeding Sciences, Hungarian University of Agricultural and Life Sciences, H-7400 Kaposvár, Hungary; kover.gyorgy@uni-mate.hu

**Keywords:** fumonisin B1, T-2 toxin, deoxynivalenol, oxidative stress, lipid peroxidation, glutathione, lipid composition, poultry

## Abstract

Fumonisin B1, T-2 toxin, and deoxynivalenol are frequently detected in feed materials. The mycotoxins induce free radical formation and, thereby, lipid peroxidation. The effects of mycotoxin exposure at the EU recommended limit (T-2/HT-2 toxin: 0.25 mg/kg; DON = 3AcDON/15-AScDON: 5 mg/kg; fumonisin B1: 20 mg/kg) and double dose (T-2/HT-2 toxin: 0.5 mg/kg, DON/3-AcDON/15-AcDON: 10 mg, and FB1: 40 mg/kg feed) were investigated during short-term (3 days) *per os* exposure in the liver of laying hens. On day 1 higher while on day 3 lower MDA concentrations were found in the low-dose group compared to the control. Fatty acid composition also changed: the proportion of monounsaturated fatty acids increased (*p* < 0.05) and the proportion of polyunsaturated fatty acids decreased by day 3. These alterations resulted in a decrease in the index of unsaturation and average fatty acid chain length. Histopathological alterations suggested that the incidence and severity of liver lesions were higher in the mycotoxin-treated laying hens, and the symptoms correlated with the fatty acid profile of total phospholipids. Overall, the findings revealed that mycotoxin exposure, even at the EU-recommended limits, induced lipid peroxidation in the liver, which led to changes in fatty acid composition, matched with tissue damage.

## 1. Introduction

Mycotoxins in feed and food are a worldwide problem because they can cause animal and human health damage and economic losses in the food and feed industry and animal husbandry. In Europe, *Fusarium* mycotoxins, such as T-2/HT-2 toxin, deoxynivalenol (DON), and fumonisin B1 (FB1), were detected in most cereals used for poultry feed, and the co-occurrence of these mycotoxins was also common [1].

Poultry species are less susceptible to fusariotoxins, so there are usually no clinical signs of exposure, but exposure can cause changes in production efficiency and reproduction. The effects of T-2 toxin and DON on laying hens include reduced feed intake [2], immunosuppression [3], and decreased egg production and quality [4]. FB1 can cause liver damage [5], reduce feed intake and efficiency [6], and alter the immune response [7]. However, the severity of the effects may vary depending on the co-occurrence of mycotoxins and the duration of exposure. The trichothecene mycotoxins, DON, and their harmful metabolites (3-AcDON, 15-AcDON) and T-2/HT-2 toxins hold significant importance due to their widespread and toxic nature [8]. Because of the epoxide ring and the 9 and 10 double bonds, they are chemically reactive compounds [9]. They induce DNA damage, protein oxidation, and lipid peroxidation, resulting changes in membrane integrity, cellular redox signaling, and antioxidant status [10]. Fumonisin B1 (FB1) inhibits sphingolipid metabolism because it is a structural analog of sphingosine. It leads to a decrease in the relative concentration of ceramide and an increase in free sphingoid bases and sphingosine-1-phosphate, which can generate oxidative stress [11]. In addition, several studies revealed the generation of oxygen free radicals and lipid peroxidation as effects of FB1 [12,13,14].

Oxidative degradation of lipids impacts the lipid and fatty acid (FA) composition of cellular membranes, with its effects closely linked to the level of FA’ unsaturation within the membranes. Previous studies showed that some *Fusarium* toxins (FB1, DON, and ZEA) have significant individual and collective impacts on the fatty acid (FA) composition of rat and rabbit liver membrane lipids [15,16]. The alterations in the concentrations of lipid mediators both inside and outside cells modify the expression and function of signaling and regulatory pathways [17]. In poultry studies, exposure to *Fusarium* mycotoxins was found to reduce the levels of markers associated with lipid peroxidation, despite evidence of oxidative stress indicated by alterations in antioxidant parameters and the expression of redox-sensitive genes [18,19], which raises interesting questions about the complex interplay between oxidative stress and the alteration of FA composition. The liver plays a crucial role in mycotoxicity experiments as it is involved in the detoxification of mycotoxins and their metabolized forms [20,21,22]. In poultry, hepatotoxic effects of T-2 toxin, DON, and FB1 have also been described. According to Yin et al. [23], the rate of apoptosis and the extent of pathological changes in hepatocytes escalated proportionally with the dose. The examination of liver tissue via histopathological analysis revealed notable alterations induced by the toxin, such as hepatocyte edema, increased volume, and heightened cytoplasmic granulation. At a DON concentration of 19.03 mg/kg, commonly observed abnormalities encompassed inflammatory infiltration, congestion, cytoplasmic vacuolation in hepatocytes, liver trabeculae disorganization, and broiler chicken necrosis [24]. Similar histological alterations were noted in the livers of birds fed FB1-contaminated feed [25]. The study aimed to investigate the interaction between T-2/HT-2 toxin, DON, and FB1 regarding the FA composition of membrane lipids, oxidative degradation of lipids, and hepatocellular integrity in the laying hen liver.

## 2. Results

### 2.1. Body and Liver Weight

The body weight (BW) and relative liver weight did not differ among groups during the study (Table 1).

### 2.2. Liver Oxidative and Antioxidant Parameters

The malondialdehyde (MDA) concentration in the liver was significantly higher in the low multi-mycotoxin dose group compared to the control group on day 1. However, it decreased significantly by day 3 (Figure 1). Among the antioxidant parameters, the amount of reduced glutathione (GSH) was significantly higher than the control as an effect of the EU-proposed dose on day 1 (Figure 1). There were no significant changes in the markers of the early stage of lipid peroxidation, as indicated by levels of conjugated dienes (CD) and conjugated trienes (CT) or the activity of glutathione peroxidase (GPx) between the experimental groups.

### 2.3. Hepatic Phospholipid Fatty Acid Profile

#### 2.3.1. Saturated Fatty Acids

The proportion of lauric acid (C12:0) decreased in all groups in an exposure time-dependent manner, while mycotoxin dose-dependent alteration was not proven (Table 2). Myristic acid (C14:0) decreased intermittently within the high-mix group on days 1 and 2. Palmitic acid (C16:0) levels were higher in both multi-mycotoxin-exposed groups on day 3 than in the control. Checking the dose-associated alterations (C16:0), a decrease was observed on days 1 and 2, while on day 3, the opposite was found. The proportion of stearic acid (C18:0) decreased with the progress of time (i.e., age in control and exposure time in the other cases) in both intoxicated groups. On days 1 and 2, increasing levels were associated with an increasing toxin load, while the opposite was found on day 3. The arachidic acid (C20:0) level decreased with time in the high-dose group and showed a dose-dependent proportional decrease on day 3. Behenic acid (C22:0) level decreased in the high-dose group at onset, but an increase was found. For lignoceric acid (C24:0), no systematic proportional change was found. The total saturated FAs showed no systematic proportional changes. However, on day 3, the saturated FA levels of both multi-mycotoxin groups exceeded those of the control group (Table A1).

#### 2.3.2. Monounsaturated Fatty Acids

The level of palmitoleic acid (C16:1 n7) decreased on day 2 corresponding with the decreasing toxin dose, while exposure time-dependent systematic change was not detected. Oleic acid (C18:1 n9) proportion markedly increased in treated groups by day 3 compared to the control. Changes in cis-vaccenic acid (C18:1 n7), gondoic, and erucic acids (C20:1 n9, C22:1 n9) were not systematic. The total monounsaturated fatty level increased markedly with exposure time and intoxication as well, reaching maxima at day 3 in both low and high groups (Table A2).

#### 2.3.3. Polyunsaturated Fatty Acids

Minor or inconsistent proportional changes were detected for linoleic (C18:2 n6), alfa-linolenic (C18:3 n3), gamma-linoleic (C18:3 n6), eicosadienoic (C20:2 n6), and eicosatrienoic (C20:3 n3) acids. Proportional changes in dihomo-gamma-linoleic acid (C20:3 n6) were not systematic according to the exposure time, but, on day 3, both mycotoxin treatments lowered their proportion. Interestingly, the Mead acid (C20:3 n9) proportion showed a similar, mycotoxin-associated decrease. The proportion of hepatic phospholipid arachidonic acid (C20:4 n6) decreased significantly due to mycotoxin loads (low and high). Similar alteration patterns were found for eicosapentaenoic (C20:5 n3), adrenic acid (C22:4 n6), docosapentaenoic (C22:5 n3), and docosahexaenoic (C22:6 n3) acids on day 3 (Table A3).

Indices calculated from the data of multiple individual FAs provided identical changes: the overall polyunsaturated FAs, the total n3 and total n6 FA levels, the unsaturation index, and the average FA chain length all declined due to intoxication by day 3, compared to control group; meanwhile, low and high treatments showed the same results (Figure 2).

### 2.4. Histopathology of the Liver

The experimental group and day-based mean total lesion score data are given in Table 2. Still, within experimental days, there was no mycotoxin dose-associated difference, while increasing total lesion scores were found with the progress of the trial. The main histopathological findings were diffuse vacuolization (Figure A1A), multifocal discrete mononuclear cell infiltration around the portal vessels (Figure A1B), and the cytoplasm of hepatocytes faintly staining around the portal vessels. Focal mononuclear cell infiltration of the parenchyma and around ductuli (Figure A1C,D) was observed, as well as hydropic degeneration of the hepatocytes. Higher total lesion sore values (all symptoms’ severity scores summarized) were found on the third day, in the high-dose group. Low-grade swelling was found in the case of the control group by day 3 (2/5) and in the low mixture multi-mycotoxin contamination (4/5), while low- (1/5) and medium-grade (1/5) swelling were found due to the high multi-mycotoxin dose. The proliferation of Kupffer cells was found in low (1/5) and low-to-medium (1/5) grades as an effect of the high multi-mycotoxin administration. Vacuolization of the cytoplasm of hepatocytes was found in low (1/5) and medium (2/5) grades in the control group on day 3. Vacuolization of the cytoplasm of hepatocytes was in low (3/5) and medium (2/5) grades within the low-dose group and medium (2/5) and high (3/5) grades for the high dose of multi-mycotoxin exposure. The focal or multifocal necrobiosis of hepatocytes occurred in a low-grade (1/5) in the control group on day 3 of the trial. Multi-mycotoxin exposure caused low-grade necrobiosis (1/5) due to the low dose, but low (2/5), medium (2/5), and high (1/5) grades were found in the group fed with high multi-mycotoxin-contaminated feed.

The numbered list of all symptoms detected to different extents/severity is given in Table A4.

### 2.5. Interrelationship between Histopathological Findings and Phospholipid Fatty Acids

The distinct individual histopathological alterations (S1–S15) detected, and the phospholipid FA profile (individual FAs and calculated FA indices) was tested for inter-relationships. Data from this test are provided in Table 3. Four symptoms of histopathological alterations provided a correlation with FAs: diffuse vacuolization; S5: blurred cell boundaries; S6: dilatated sinusoids; and S15: faint cytoplasm staining. The S1 and S5 symptoms provided negative and positive correlations with alpha-linolenic acid, respectively, while S5 also had negative and positive correlations with stearic acid (C18:0) and lignoceric acid (C24:0). S6 had only one positive correlate, myristic acid (C14:0). At the same time, S15 was strongly and negatively related to lauric acid proportion (C12:0). The total lesion score (cumulative scores in the full S1–S15 range) showed a slight positive correlation with oleic acid (C18:1 n9) and a slight negative correlation with dihomo-gamma-linolenic and arachidonic acids (C20:3 n6 and C20:4 n6).

## 3. Discussion

The combined exposure of laying hens to T-2/HT-2 toxin, DON/3-AcDON/15-AcDON, and FB1 at the doses of the EU limits and at double dose led to the formation of reactive oxygen species (ROS) in the liver on the first day of the experiment. This ROS formation subsequently induced lipid peroxidation, increasing MDA levels, a lipid peroxidation marker [26]. In response, an increase in GSH content activated the antioxidant defense system. The same result was obtained in a previous experiment where equal multi-mycotoxin exposure activated the glutathione redox system and its coding genes due to oxidative stress [27]. By the third day of the present study, the MDA concentration decreased in the same dose group. This reduction was observed in an earlier study [19] using individual T-2 toxins and DON at the double dose of the EU-proposed limit. This decrease was, with the highest probability, caused by a change in the lipid–FA composition of liver tissue due to mycotoxin exposure: the ratio of monounsaturated FAs increased, while the proportion of polyunsaturated FAs decreased. This alteration was also described as an effect of FB1 on rabbits [15]. The proportional depletions in omega-3 and omega-6 polyunsaturated FAs due to multi-mycotoxin exposure were accompanied by notable decreases in unsaturation index and average chain length. These changes affect the sensitivity of lipids to oxidation and lipid peroxidation that occur as a result of ROS [28].

The liver is the primary site of detoxification for many mycotoxins, including trichothecenes [29]. The FB1 inhibits ceramide synthase, leading to sphinganine accumulation and disrupting cellular functions [30]. FB1 exposure can also induce oxidative stress, leading to the generation of ROS and oxidative damage of cellular components by releasing pro-inflammatory cytokines [31]. Individual mycotoxin exposure increased oxidative stress in the dose of EU-proposed limits, resulting in the peroxidation of lipids as observed in short-term exposure to high individual doses of T-2 toxin in laying hens [32]. This result allows us to suggest the same effect for multi-mycotoxin exposure. On the third day of the study, significant alterations in FA proportions were observed concurrently with the cotoxin dose. The proportion of unsaturated FAs and the average chain length decreased in all intoxicated groups compared to the control. This observation aligns with findings from a previous study with rats [16], where the fatty acid profile of hepatocellular membrane lipids demonstrated decreased unsaturation, primarily influenced by DON, and, to a lesser extent, by zearalenone. Under the influence of the applied doses of the mycotoxin mixture, the antioxidant system cannot fully inhibit the formation of oxygen free radicals; therefore, lipid peroxidation processes are induced, primarily affecting polyunsaturated FAs, so their amount decreased within the total FA pool. The histopathological results revealed that the incidence and severity of lesions showed no significant variation among the experimental groups in any sampling. Histological examination showed the swelling of cells belonging to the mononuclear phagocyte system (MPS) and Kupffer cells within the hepatic tissue [33]. The slight swelling of MPS cells and the slight degree of vacuolation of hepatocytes in the untreated control groups could connect with the high nutrient supply during the rearing period [34]. The higher incidence and severity of these lesions, the necrobiosis of hepatocytes, and the proliferation of MPS cells are related to mycotoxin exposure. The histopathological evaluation revealed a mild status, likely attributed to the short duration of the experiment and the low doses of mycotoxins administered. However, these lesions could negatively affect the production and health status of affected animals and cause immunosuppression [35]. The immunotoxic effects of the tested mycotoxins may make laying hens more sensitive to numerous infectious diseases. In contrast to our study, longer exposure periods in other investigations likely played a significant role in the development of liver alterations. Histological observations in birds exposed to T-2 toxin for 21 days and DON for 7 days revealed swollen and necrotic hepatocytes [24,36].

It is worth mentioning that the impact of mycotoxin-induced changes in FA composition and on lipid peroxidation markers may vary depending on the specific mycotoxin, its concentration, the duration of exposure, and the antioxidant capacity of the organism. Other factors, such as the balance between pro-oxidants and antioxidants and the activity of antioxidant enzymes, can also influence the overall oxidative status and lipid peroxidation.

## 4. Conclusions

Combined *Fusarium* mycotoxin exposure caused marked oxidative stress-inducing effects even at the EU-recommended limit values for the respective individual mycotoxins. It suggested that EU recommendation limit levels should be modified for multi-mycotoxin exposure, which often occurs in poultry practice. Because of the mycotoxin-induced oxidative stress, the intensity of lipid peroxidation processes decreased by the third day of the study, which relates to the changes in the FA composition of the liver phospholipids. Differences in the phospholipid FA profile, particularly a decrease in the proportion of polyunsaturated FAs, cause tissue damage, as was proven by histopathological examination, coupled with a loss of membrane integrity.

## 5. Materials and Methods

### 5.1. Experimental Setup and Conditions

The study involved 60 Tetra SL laying hens as an in vivo model, aged 49 weeks, with an average daily egg production of 90%. The animals (n = 60) were divided into three groups, control (n = 18), low mixture (n = 18), and high mixture (n = 18) groups, after a 3-day acclimatization period. The birds in each experimental group were divided into two subgroups (n = 9) and placed in the experimental room. Following a 12 h period of feed deprivation, a 3-day feeding trial commenced. On day 0 of the study, two animals per experimental group (n = 6) were sampled as absolute controls. Six animals from each group were sampled on the trial’s first, second, and third day. The laying hens were fed ad libitum and had free access to drinking water throughout the study. The nutrient content of the poultry diet was specified: 16.10% crude protein, 89.20% dry matter, 5.50% crude fiber, 4.12% calcium, 2.50% ether extract, 0.38% methionine, 0.79% lysine, 0.71% methionine + cysteine, 0.48% phosphorus (available), 0.17% sodium, and 11.97 MJ/kg M.E. The animals were kept in deep litter under a lighting schedule of 12 h light and 12 h dark. The laying hens were exposed to low and high doses of multi-mycotoxin contamination in their feed. The low mixture consisted of T-2/HT-toxin (0.25 mg/kg feed), DON/3-AcDON/15-AcDON (5 mg/kg feed), and FB1 (20 mg/kg feed), while the high mixture comprised T-2/HT-2 toxin (0.5 mg/kg feed), DON/3-AcDON/15-AcDON (10 mg/kg feed), and FB1 (40 mg/kg feed). The European Union (EU) has established regulatory limits for mycotoxin contamination in feed [37]. The “low dose” used in the experiment was the proposed limit of the individual mycotoxins in poultry feeds. Using the dose of the EU proposed limits, this study aimed to assess the effects of mycotoxin contamination that could be encountered in practice, such as through feed ingredients that may have this level of contamination due to environmental factors or improper storage conditions [1]. The “high dose” was designed to induce measurable changes in a short-term experiment.

*Post-mortem* liver samples, which are involved in the metabolism of mycotoxins, were taken. Liver samples were collected 24, 48, and 72 h after the start of mycotoxin exposure. The samples were randomly obtained from six animals of each group, rinsed with ice-cold isotonic saline, and then stored at −70 °C until analysis.

### 5.2. Mycotoxin Production and Analysis

The feed was mixed with sterilized corn grain artificially infected with toxinogenic *Fusarium sporotrichioides* (NRRL 3299) fort T-2/HT-2 toxin, *Fusarium graminearum* (NRRL 5883) for DON/3-AcDON/15-AcDON, and using *Fusarium verticillioides* (MRC 826) for FB1. These fungi are known producers of these specific mycotoxins, and their cultivation on corn substrate mimics natural contamination.

The mycotoxin content of the experimental feed was measured as follows. For the extraction, 1.00 ± 0.01 g feed sample was placed in a 50 mL polypropylene centrifuge tube, and 4 mL of extraction solvent (acetonitrile/water/formic acid: 49/49/2 *v*/*v*/*v*) was added to it. The tubes were then shaken for 15 min using a horizontal rotary shaker (320 rpm). A mixture of 0.8 g anhydrous MgSO_4_ and 0.2 g NaCl powder was added and was vortexed immediately for 60 s. The resulting mixture was shaken for another 15 min, then centrifuged for 5 min (4000 rpm). Next, 250 µL of the supernatant acetonitrile phase was diluted to 1 mL with de-ionized water. After filtration with a 0.22 µm syringe filter, 10 µL of the prepared sample was injected into the LCMS system. The concentrations of mycotoxins in the prepared samples were measured using a Shimadzu 2020 LCMS system equipped with an electrospray ion source (ESI) operating in positive and negative ion modes. For optimal chromatographic separation, an XB-C18 Kinetex analytical column (100 × 2.1 mm, 2.6 µm; Phenomenex) was used at a flow rate of 0.3 mL/min. The injected sample volume was 10 µL, and the column temperature was maintained at 40 °C. Gradient elution was performed employing eluents A (0.1% formic acid + 0.005 M ammonium formate) and B (0.1% formic acid in acetonitrile), using the following gradient program: the rate of eluent B started from 10%, which was linearly increased in 8 min to 100%; the column was then washed with pure eluent B for 3 min; the starting conditions were reestablished linearly decreasing the rate of eluent B in 1 min; and then the column was re-equilibrated for 3 min with 10% eluent B.

Table 4 presents the concentrations of T-2/HT-2 toxin, DON/3-AcDON/15-AcDON, and FB1 in the experimental feeds.

The determined recovery values for DON and T-2 were 109 ± 3% and 103 ± 5%, respectively. The apparent difference (measured vs. EU-recommended limit) for DON and T-2 toxins may be caused by artificial contamination. The corn-fungal culture may be somewhat inhomogeneous, and by diluting a very concentrated stock to the ready-to-feed diet, concentration differences may develop between predicted and measured values.

### 5.3. Measurement of the Antioxidant and Oxidative Markers

The samples were thawed at room temperature and homogenized in physiological saline (0.65% (*w*/*v*) NaCl) at a 1:9 ratio for biochemical analyses. A 1:9 ratio of the native homogenates was used to determine the MDA content, while the markers of the glutathione redox system were assessed using the supernatant obtained after centrifuging the homogenates (10,000× *g*, 3 min, 4 °C), which corresponds to the microsomal fraction.

CD and CT were determined from the absorption spectra (CD: 232 nm; CT: 268 nm) after the extraction of the lipid content of the samples in 2,2,4-trimethylpentane (Sigma, St. Louis, MO, USA) [38]. MDA concentration was defined by complex formation with 2-thiobarbituric acid in an acidic medium at high temperature (Sigma, St. Louis, MO, USA) according to the method by Fawaeir et al. [39]. Then, 10% (*w*/*v*) trichloroacetic acid (Carlo Erba, Rodano, Italy) was used to adjust the acidic medium. The reaction time was 20 min at 100 °C. After cooling and centrifugation (2500 rpm, 4 °C), the absorbance was measured from the supernatant against the reagent blank at a spectrum of 535 nm [40].

The reduced glutathione concentration of the samples was determined by the method of Sedlak and Lindsay [41]. The method is based on the color complexation reaction of the free SH group of glutathione with a sulfhydryl-reactive compound. The protein content of the samples was precipitated with 10% (*w*/*v*) trichloroacetic acid (Carlo Erba, Rodano, Italy), and measurements were made from the supernatant after centrifugation (10,000× *g*, 3 min, 4 °C). The concentration of GSH can be determined by measuring the absorbance at 412.

The basis of determining GPx activity is that GSH is oxidized to glutathione disulfide by GPx in the presence of ROS [42]. After 10 min incubation at room temperature (25 ± 2 °C), the reaction was stopped by protein precipitation with 10% (*w*/*v*) trichloroacetic acid (Carlo Erba, Rodano, Italy). The decrease in the amount of GSH was determined by measuring the absorbance of the complex formed with 5,5-dithiobis-(2-nitrobenzoic acid) (Sigma, St. Louis, MO, USA) at a wavelength of 412 nm. Enzyme activity is expressed in units of 1 nmol GSH oxidation per minute in the system used at 25 °C.

GSH content and GPx activity were determined relative to the protein content of the supernatant fraction using the Folin–Ciocalteu phenol reagent [43].

### 5.4. Lipid Analysis

Liver samples (ca. 100 mg, thawed just before extraction) were homogenized in 20-fold chloroform–methanol (2:1 vol–vol) (IKA Ultra Turrax, T18, IKA, Staufen, Germany), and total lipid content was extracted according to Folch et al. [44]. High-purity solvents (99.5% purity or higher, Merck, Schnelldorf, Germany) were utilized, with the addition of 0.01% *w*/*v* butylated hydroxytoluene to inhibit FA oxidation. In the frame of lipid fractionation, extracted total lipids were transferred to glass chromatographic columns containing 300 mg silica gel (200–425 mesh, Merck #236772) for 10 mg of lipids, according to Leray et al. [45]. Neutral lipids were separated using 10 mL chloroform, followed by the addition of 15 mL acetone–methanol (9:1 vol–vol) for the specified fat quantity. Total phospholipids were eluted using 10 mL pure methanol. This latter fraction was evaporated under a nitrogen stream and transmethylated with Christie’s base-catalyzed NaOCH_3_ method [46].

FA methyl-esters were extracted using 300 μL ultrapure n-hexane. Gas chromatography was performed using an AOC 20i automatic injector connected to a Shimadzu Nexis 2030 system (Kyoto, Japan) equipped with a Phenomenex Zebron ZB-WAXplus capillary GC column (30 m × 0.25 mm ID, 0.25 μm film, Phenomenex Inc., Torrance, CA, USA) and a flame ionization detector (FID). Operating conditions included an injector temperature of 220 °C, detector temperature of 250 °C, and a helium flow rate of 28 cm/s. The oven temperature was programmed as follows: starting from 60 °C with a 2 min hold, increasing to 150 °C, then from 150 to 180 °C at a rate of 2 °C/min with a 10 min hold at 180 °C, and finally from 180 to 220 °C at a rate of 2 °C/min with a 16 min hold at 220 °C (total duration: 74 min). Nitrogen was used as the makeup gas. Data analysis was performed using LabSolutions 5.93 software, utilizing the PostRun module (Shimadzu, Kyoto, Japan) with manual peak integration. FAs were identified based on the retention times of an external CRM standard (Supelco 37 Component FAME Mix, Merck-Sigma Aldrich, CRM47885). C22:4 n6 and C22:5 n6 standards were purchased from Merck (cat. no.: D3534) and Larodan (Solna, Sweden, cat. no.: 10-2265-4), respectively. The results for FAs were presented as the percentage of total FA–methyl esters by weight.

### 5.5. Histological Sample Preparation

Tissue samples were preserved in 10% neutral buffered formalin and subsequently embedded into paraffin. Slides of 5 μm thickness were prepared using a microtome (Reichert RF-800 model, Precisionary, Natick, MA, USA) and stained with hematoxylin–eosin for subsequent light microscopic analysis.

The main pathological alterations (referred to as symptoms, S) were described and scored according to their occurrence and severity: 0 = no alteration; 1 = slight/small scale/few; 2 = medium degree/medium scale/medium number; 3 = pronounced/extensive/numerous.

### 5.6. Data Analysis

Statistical analysis was performed using GraphPad Prism 6.07 software from GraphPad Software in San Diego. Data are expressed as mean ± standard deviation (SD). Group and day means were compared with a two-way ANOVA, followed by the Tukey post hoc test to detect inter-group differences. Linear regression was employed to examine the dose–response relationship. Statistical significance between groups was determined at *p* < 0.05. Histopathological assessment involved scoring severity from 0 to 3 based on symptom severity. Histopathological alterations were statistically evaluated with a two-way ANOVA, the two factors being exposure time (day) and the second mycotoxin dose. The continuous PL FA and the ranked histopathology data were tested for correlation using Spearman’s rho rank correlation method. The analyses mentioned were conducted using IBM SPSS 20.0 (2012) software.

### 5.7. Ethical Allowance

The study adhered to the guidelines outlined in the European Communities Council Directive (86/609 EEC). It sets guidelines for protecting animals used for scientific purposes, ensuring their welfare and ethical treatment during experiments.

## Figures and Tables

**Figure 1 toxins-16-00179-f001:**
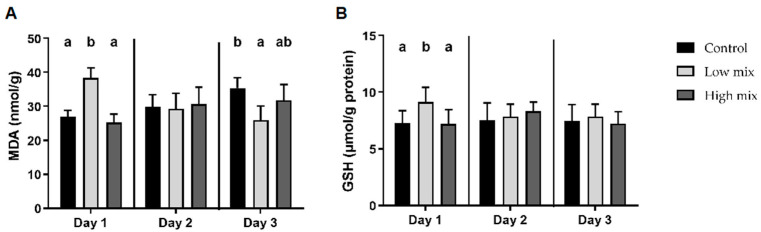
Malondialdehyde (**A**) and reduced glutathione levels (**B**) in laying hens’ liver (mean ± S.D.; n = 6). a, b: different superscripts within the same column indicate significant differences (*p* < 0.05). Low mix: T-2/HT-2 toxin (0.25 mg/kg); DON/3-AcDON/15-AcDON (5 mg/kg); FB1 (20 mg/kg feed); High mix: T-2/HT-2 toxin (0.5 mg/kg); DON/3-AcDON/15-AcDON (10 mg/kg); FB1 (40 mg/kg feed).

**Figure 2 toxins-16-00179-f002:**
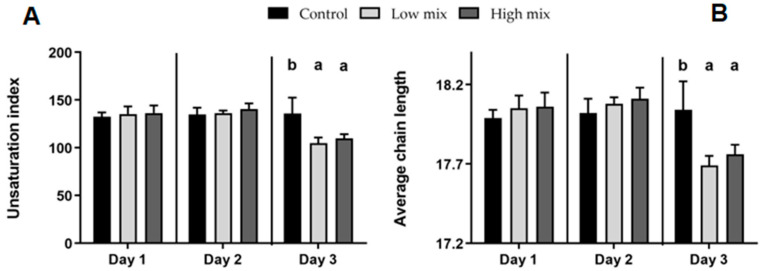
Calculated fatty acid indices, the unsaturation index (the number of double bonds in 100 fatty acyl chains) (**A**), and the average fatty acid chain length (**B**). a, b: different letters within the same column indicate significant differences (*p* < 0.05). Low mix: T-2/HT-2 toxin (0.25 mg/kg); DON/3-AcDON/15-AcDON (5 mg/kg); FB1 (20 mg/kg feed); High mix: T-2/HT-2 toxin (0.5 mg/kg); DON/3-AcDON/15-AcDON (10 mg/kg); FB1 (40 mg/kg feed).

**Table 1 toxins-16-00179-t001:** Weight data of laying hens in control and multi-mycotoxin exposure groups (mean ± S.D.; n = 6).

	Exp. Day	Control			Low Mix			High Mix		
Body weight (g)	0	1645.8	±	317.6	1645.8	±	317.6	1645.8	±	317.6
	1	1590.8	±	209.5	1537.5	±	138.9	1709.2	±	158.1
	2	1699.5	±	73.4	1597.0	±	120.3	1615.8	±	123.7
	3	1652.8	±	172.6	1626.2	±	77.2	1677.3	±	105.2
Relative liver weight (% of BW)	0	1.54	±	0.14	1.54	±	0.14	1.54	±	0.14
	1	1.79	±	0.14	1.74	±	0.25	1.89	±	0.16
	2	1.83	±	0.14	1.90	±	0.32	1.62	±	0.18
	3	1.92	±	0.34	1.95	±	0.48	1.77	±	0.20

Low mix: T-2/HT-2 toxin (0.25 mg); DON/3-AcDON/15-AcDON (5 mg); FB1 (20 mg/kg feed); High mix: T-2/HT-2 toxin (0.5 mg); DON/3-AcDON/15-AcDON (10 mg); FB1 (40 mg/kg feed).

**Table 2 toxins-16-00179-t002:** The total lesion score (S1–S15: please see for details Table A4) group means according to experimental days and mycotoxin doses (mean ± S.D.).

Exp. Time	Control	Low Mix	High Mix
*Liver*			
Day 1	2.17 ± 1.60 ab	1.83 ± 1.47 a	2.17 ± 0.75 ab
Day 2	1.67 ± 1.21 a	2.17 ± 1.47 ab	1.67 ± 0.81 a
Day 3	4.00 ± 2.28 b	4.00 ± 1.67 b	3.50 ± 1.37 b

a,b: different letters within the same column mean significant difference (*p* < 0.05). Low mix: T-2/HT-2 toxin (0.25 mg/kg); DON/3-AcDON/15-AcDON (5 mg/kg); FB1 (20 mg/kg feed); High mix: T-2/HT-2 toxin (0.5 mg/kg); DON/3-AcDON/15-AcDON (10 mg/kg); FB1 (40 mg/kg feed).

**Table 3 toxins-16-00179-t003:** The correlation between phospholipid fatty acid profile and histopathological symptoms of the liver.

Symptom/FA		C12:0	C14:0	C18:0	C18:1 n9c	C18:3 n3	C20:3 n6	C20:4 n6	C22:5 n6	C24:0
**S1**	Correlation Coefficient					−0.57				
	*Sig. (2-tailed)*					0.04				
**S5**	Correlation Coefficient			−0.50		0.49				0.58
	*Sig. (2-tailed)*			0.04		0.04				0.02
**S6**	Correlation Coefficient		0.46						−0.46	
	*Sig. (2-tailed)*		0.04						0.04	
**S15**	Correlation Coefficient	−0.91								
	*Sig. (2-tailed)*	0.03								
**total lesion score**	Correlation Coefficient				0.38		−0.30	−0.38		
	*Sig. (2-tailed)*				0.00		0.02	0.00		

**Table 4 toxins-16-00179-t004:** Measured mycotoxin content of the feed (mg kg^−1^ feed).

Group	T-2/HT-2	DON/3-AcDON/15-AcDON	FB1
Control	<0.01	<0.01	<0.01
Low mix	0.13/-	0.67/0.56/0.01	20.00
High mix	0.29/-	2.70/0.89/0.05	40.30

## Data Availability

The raw data supporting the conclusions of this manuscript will be made available by the authors, without undue reservation, to any qualified researcher.

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
