# Peer review of "The Effect of Combined Exposure of Fusarium Mycotoxins on Lipid Peroxidation, Antioxidant Defense, Fatty Acid Profile, and Histopathology in Laying Hens’ Liver"

_toxins, 2024, doi:10.3390/toxins16040179_

Round 1
Reviewer 1 Report
Comments and Suggestions for Authors
In this study the Authors investigated the effects of mycotoxin exposure at the EU recommended limit (T-2/HT-2 toxin: 0.25 mg/kg; DON=3AcDON/15-AScDON: 5 mg/kg; fumonisin B1: 20 mg/kg) and twice dose (T-2/HT-2 toxin: 0.5 mg/kg, DON/3-AcDON/15- AcDON: 10 mg, and FB1:40 mg/kg feed) were investigated during short-term (3 days) per os exposure in laying hens’ liver. I think the toxin addition time in laying hens feed is too short and does not have practical application. Lack of description of the execution method for experimental chickens. In addition, the research content in this experiment is similar to previous research, lacking research innovation and depth.
Comments on the Quality of English LanguageThe writing needs to be polished.
Reviewer 2 Report
Comments and Suggestions for Authors
Overall, the paper is well-written and the evidence supports the conclusions. The only advice is more aesthetic than scientific:
1) I found the use of letters in figures 1 and 2 to designate significance somewhat confusing since the panels also have the same letters. My suggestion is to change the panel letters for a to A and from b to B.
2). The figures showing liver damage are visually impressive. However, instead of having the figures shown separately, I'd put them together in one figure as separate panels. It makes it easier for the reader to to make comparisons, instead of switching from figure to figure.
Reviewer 3 Report
Comments and Suggestions for Authors
The paper deals with how exposure to Fusarium mycotoxins influences lipid peroxidative and antioxidant defense, fatty acid profile and liver histopathology in laying hens. The introduction provides bibliographic data regarding the effect of Fusarium mycotoxins on the animal body, but more references to laying hens would be necessary.
The study and analysis methods are described in detail and can be reproduced. The results are clearly presented and are supported by experimental data.
In the Discussions chapter, the presented conclusions are recommended to be supported/compared by a larger number of data from relevant references
Reviewer 4 Report
Comments and Suggestions for Authors
1. Abstract, rather than giving qualitative data (just saying increase or decrease) authors should provide quantitative data in the abstract.
2. Author should provide conclusive statement in abstract too.
3. Keywords: Authors can choose keyword more wisely, keywords such as 'laying hen' and 'lipid compositions' should be replaced.
4. In Introduction, authors have mainly focused on the toxicity and toxic effect of fusarium mycotoxins. Author should focus on the main problem, objective and applicability of the current study.
5. Section 5.3, should be explained in detail, how much tissue and buffer used for extraction and how much MDA were added, and what was MDA concentration?
6. Overall methodology section is too weak, it should be revised thoroughly and provide detailed methods for each section.
7. It is recommended to carry out MPO assay for neutrophil infiltration.
Comments on the Quality of English LanguageModerate English editing required
Reviewer 5 Report
Comments and Suggestions for Authors
In materials and methods in the chapter Experimental setup and conditions, a fortification of the fodder is made with mycotoxin mixture in low and double doses, but which are found only in the case of fumonisin B1. The question is whether the quantity below half found in the mycotoxins analysis of low and high mixture of mycotoxin is due to synergistic actions or a low recovery rate, which was not mentioned in chapter Mycotoxin production and analysis. Recovery rate could also give some explanations.
In Discussions the results are compared with literature but with paper quite old as half of the bibliographic references are articles with an age of over 10 years, which should be reviewed. more are miswritten in the text such as the reference to the values of mycotoxins in the EU referred to in Article 35 and it is actually found in number 32.
Comments on the Quality of English LanguageMinor editing of English required
Round 2
Reviewer 1 Report
Comments and Suggestions for Authors
The paper can be accepted
Author Response
Thank you very much for your reply.
Reviewer 4 Report
Comments and Suggestions for Authors
All the required corrections has been made. Manuscript can be accepted
Comments on the Quality of English LanguageMinor English Editing required
Author Response
Thank you for your comments. We have improved the English quality of the article as you requested.
Reviewer 5 Report
Comments and Suggestions for Authors
I accept some of the answers, however, a recovery rate is necessary to complete the explanation regarding the low concentrations of mycotoxins. As far as concerns discussion correlated with the bibliography in now up to date.
Comments on the Quality of English LanguageMinor editing of English required
Author Response
Thank you for your comments. The recovery values for DON and T-2 were determined and included in the manuscript. We have improved the English quality of the article as you requested.